

# Effects of whole-body vibrations on neuromuscular fatigue: a study with sets of different durations

Miloš Kalc[1,2], Ramona Ritzmann[3] and Vojko Strojnik[2]

[1] Faculty of Medicine, Institute of Sports Medicine, University of Maribor, Maribor, Slovenia
[2] Faculty of Sport, Institute of Kinesiology, University of Ljubljana, Ljubljana, Slovenia
[3] Department of Biomechanics, Praxisklinik Rennbahn AG, Muttenz, Switzerland

## ABSTRACT

**Background.** Whole body vibrations have been used as an exercise modality or as a tool to study neuromuscular integration. There is increasing evidence that longer WBV exposures (up to 10 minutes) induce an acute impairment in neuromuscular function. However, the magnitude and origin of WBV induced fatigue is poorly understood.

**Purpose.** The study aimed to investigate the magnitude and origin of neuromuscular fatigue induced by half-squat long-exposure whole-body vibration intervention (WBV) with sets of different duration and compare it to non-vibration (SHAM) conditions.

**Methods.** Ten young, recreationally trained adults participated in six fatiguing trials, each consisting of maintaining a squatting position for several sets of the duration of 30, 60 or 180 seconds. The static squatting was superimposed with vibrations ($WBV_{30}$, $WBV_{60}$, $WBV_{180}$) or without vibrations ($SHAM_{30}$, $SHAM_{60}$, $SHAM_{180}$) for a total exercise exposure of 9-minutes in each trial. Maximum voluntary contraction (MVC), level of voluntary activation (%VA), low- ($T_{20}$) and high-frequency ($T_{100}$) doublets, low-to-high-frequency fatigue ratio ($T_{20/100}$) and single twitch peak torque ($TW_{PT}$) were assessed before, immediately after, then 15 and 30 minutes after each fatiguing protocol.

**Result.** Inferential statistics using RM ANOVA and post hoc tests revealed statistically significant declines from baseline values in MVC, $T_{20}$, $T_{100}$, $T_{20/100}$ and $TW_{PT}$ in all trials, but not in %VA. No significant differences were found between WBV and SHAM conditions.

**Conclusion.** Our findings suggest that the origin of fatigue induced by WBV is not significantly different compared to control conditions without vibrations. The lack of significant differences in %VA and the significant decline in other assessed parameters suggest that fatiguing protocols used in this study induced peripheral fatigue of a similar magnitude in all trials.

Corresponding author
Miloš Kalc, milos.kalc@ism-mb.si

## INTRODUCTION

Whole body vibration (WBV) transfers sinusoidal oscillations into the human body, which inspired the use of this physical modality both as a tool to study the sensorimotor integration

of the neuromuscular system and as an intervention stimulus with beneficial effects on performance (*Rittweger, 2010*). Early studies have suggested that short single sessions of WBV of 3- to 5-minutes duration in a squat position immediately increase neuromuscular performance (*Bosco et al., 2000*; *Cardinale & Bosco, 2003*), maximal voluntary contraction (MVC), jump performance and myoelectric activity (*Alam, Khan & Farooq, 2018*). The acute increase in neuromuscular performance after vibration is referred to as 'post-activation potentiation' (PAP) for short-lasting enhancements (less than 1 min) and as 'post-activation performance enhancement' (PAPE) for more extended performance enhancement periods lasting up to several hours (*Blazevich & Babault, 2019*). Both phenomena are related to vibration-induced changes in the neuronal control of the affected skeletal muscles that encompass a facilitated central drive (*Mileva, Bowtell & Kossev, 2009*; *Krause et al., 2017*) concomitant with modified reflexive activation at the spinal level (*Rittweger, Beller & Felsenberg, 2000*; *Ritzmann et al., 2018*) persistent over a period of 15 min after vibration exposure (*Krause et al., 2016*; *Ritzmann et al., 2018*).

In everyday practice, therapists and practitioners promote longer WBV exposures (up to 10 min), although the effects of such exercise modalities are mostly unknown (*Torvinen et al., 2002*; *Zory et al., 2013*). By increasing the WBV stimuli duration up to a cumulative total of 4 to 10 min, it has been suggested that WBV may acutely induce fatigue rather than potentiation (*Torvinen et al., 2002*; *De Ruiter et al., 2003*; *Erskine et al., 2007*; *Rittweger, 2010*; *Zory et al., 2013*). For example, *Torvinen et al. (2002)* and *De Ruiter et al. (2003)* observed an immediate decrease of MVC after a $10 \times 1$-minute WBV intervention. However, no changes in MVC were observed in the control condition without vibrations. Even though various studies have reported a fatigue-induced drop in neuromuscular performance, there have been contradictory findings regarding the underlying mechanisms which favour either a central or peripheral origin. Several authors investigated the effect of WBV on central fatigue (*Jordan et al., 2010*; *Maffiuletti et al., 2013*; *Zory et al., 2013*) and were unable to find any difference in the level of voluntary activation (%VA) between interventions with and without vibrations. To the best of our knowledge, the force-frequency fatigue-related mechanisms of WBV-induced peripheral fatigue have not been studied. By comparing the ratio of the electrically induced mechanical responses using low-frequency (below fusion frequency –20 Hz) and high-frequency (above fusion frequency –100 Hz) paired supramaximal electrical stimuli, peripheral fatigue can be subdivided into low- and high-frequency (*Edwards, 2008*; *Millet et al., 2011*). Analogous exercise-induced fatigue studies have demonstrated that prolonging exercise stimuli can shift the peripheral fatiguing mechanism towards low-frequency fatigue (*Millet & Lepers, 2004*; *Tomazin et al., 2012*).

To better understand the intervention stimuli induced by WBV, it is crucial to establish which fatiguing mechanisms occur after a single session of WBV, and how different vibration parameters affect the magnitude and origin of neuromuscular fatigue. The scientific and practitioner choices for WBV intervention are motivated by achieving high superimposed effects throughout WBV to trigger physiological and neuromuscular adaptations and thus, WBV parameters are combined accordingly (*Abercromby et al.,*
_2007_; _Ritzmann, Gollhofer & Kramer, 2013_). Electromyography studies suggest that side-alternating vibration exposure driven by high amplitude and frequency cause the highest activation intensities in distal and proximal leg musculature (_Abercromby et al., 2007_; _Rittweger, 2010_; _Ritzmann, Gollhofer & Kramer, 2013_). In addition to vibration-associated attributes, and in an analogy to strength training, the training load is mainly determined by intensity and volume (_Baechle & Earle, 2008_). Therefore, volume is subdivided into number of set and repetitions with defined set duration (_Campbell et al., 2017_). In a similar manner, vibration amplitude and frequency define the training intensity in WBV interventions. However, to the best of our knowledge, there is a lack of studies investigating how WBV intervention volume (set numbers and set duration) affects the occurrence of neuromuscular fatigue.

Therefore, the aim of the present study was to investigate the magnitude and origin of neuromuscular fatigue induced by long-exposure half-squat whole-body vibration intervention (WBV) with sets of different duration and compare it with non-vibration (SHAM) conditions. Thus, and with reference to, previous research involving long-exposure WBV induced fatigue (_Erskine et al., 2007_; _Zory et al., 2013_) we selected a long (cumulative exercise time of 9 min) static WBV fatiguing intervention divided into sets of different duration (30 s, 60 s or 180 s). In a series of MVC paradigms, we applied different peripheral nerve stimulation techniques, allowing us to distinguish the source of fatigue. We hypothesised that WBV exercise interventions would cause higher magnitudes of fatigue compared to non-vibration intervention (_Erskine et al., 2007_; _Zory et al., 2013_). We expected that fatigue magnitude would be dependent on the duration of exposure and would increase with set-duration. We hypothesised that predominantly peripheral, rather than central fatiguing mechanisms, would be causally involved (_Jordan et al., 2010_; _Maffiuletti et al., 2013_; _Zory et al., 2013_).

## MATERIALS & METHODS

### Study design

In a cross-over repeated measures design, each subject performed three different fatiguing exercise interventions with WBV and three exercise interventions in a sham condition without WBW (SHAM) to determinate the effect of WBV (Fig. 1A). Each intervention comprised a cumulative exercise period with a duration of 9 min divided into different sets (either $18 \times 30$ s or $9 \times 60$ s or $3 \times 180$ s) with 120 s rest between sets (Fig. 1A). The exercise interventions were performed on an activated vibration platform ($WBV_{30}$, $WBV_{60}$, $WBV_{180}$) and three on an inactive vibration platform ($SHAM_{30}$, $SHAM_{60}$, $SHAM_{180}$). Each intervention was executed on different visits with at least seven days rest in-between. The order was randomised. The subjects were not permitted to undertake explosive strength training or fatiguing workouts for 48 h before each measuring day, in order to eliminate side-effects. The study design, materials and neuromuscular assessments are available for reference in protocols.io (dx.doi.org/10.17504/protocols.io.beadjaa6).

Neuromuscular assessment in the resting position was performed at $t_0$ (baseline) prior to exercise intervention. The assessment consisted maximum voluntary contraction (MVC)

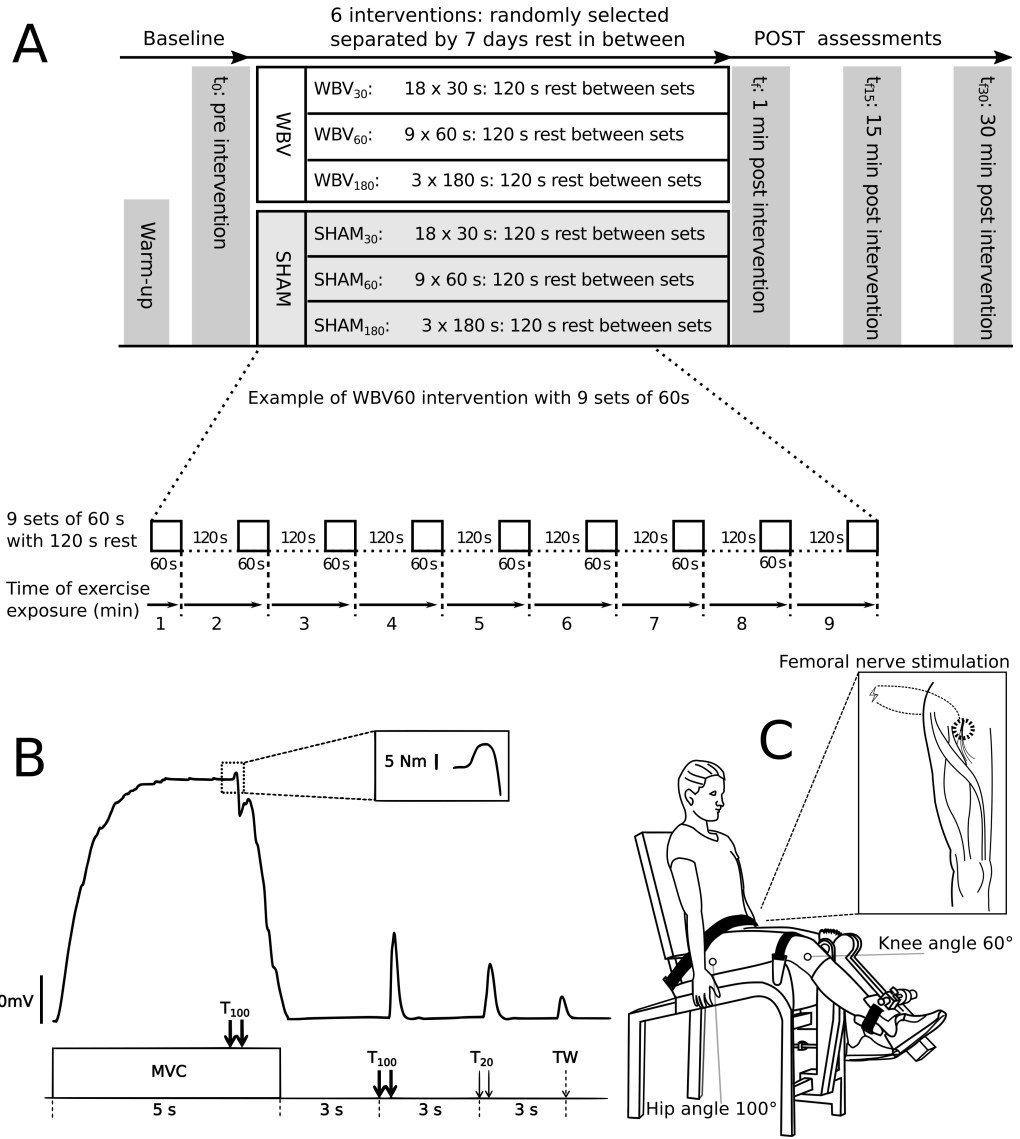

**Figure 1 Experimental design and settings.** (A) Experimental design comprising the fatiguing protocols for all six visits and the timeline of each visit. Neuromuscular function was assessed before ($t_0$), immediately after ($t_f$), 15 ($t_{f15}$) and 30 ($t_{f30}$) minutes after vibration intervention. An expanded view of exercise exposure representing the $WBV_{60}$ protocol (nine sets of 60 s of vibration exercise with 120 s rest between sets) is presented in detail. (B) Example of a torque signal from the neuromuscular testing procedure. An expanded view of an interpolated twitch is presented in the dotted box. The neuromuscular testing procedure comprised MVC of the quadriceps muscle combined with different electrical stimulation methods to assess the level of voluntary activation - %VA (via the interpolated double twitch technique), quadriceps twitch torques in response to paired electrical stimuli at 100 Hz ($T_{100}$) and at 20 Hz ($T_{20}$), as well as single twitch (TW). (C) Schematic of the position of the subject during the neuromuscular assessment. An expanded view of the femoral nerve stimulation point in the popliteal fossa is presented within the box.

of the knee extensors, interpolated with a high frequency ($T_{MVC}$) twitch (10 ms interstimuli interval), followed 3 s later by a 100 Hz doublet ($T_{100}$), followed 3 s later by a 20 Hz (50 ms interstimuli interval) doublet ($T_{20}$), and 3 s later by a potentiated single twitch (TW).

The assessment procedure was executed according to (*Millet et al., 2011*) and repeated at 1 min ($t_f$), as well as at 15 ($t_{f15}$) and 30 min ($t_{f30}$) after the final 9-minute intervention. All neuromuscular assessments were performed on the right leg.

## Subjects

Ten healthy subjects (6 men and 4 women; age: 21.1 ± 1.41 years, mass: 77.8 ± 11.73 kg, BMI: 22.9 ± 1.25) volunteered to participate in the study. All subjects were recreationally trained athletes, participating in moderate endurance and strength training activities 3 times per week. Exclusion criteria were acute injuries in the upper or lower extremities, locomotor dysfunctions, pregnancy, cardiovascular or neurological conditions. All subjects signed the written informed consent and the study was approved by the Ethics Committee of the Faculty of Sport of the University of Ljubljana 975/2017 and conducted according to the Declaration of Helsinki II.

The sample size was estimated by means of a power analysis aiming to detect large effect sizes ($f = 1.2$; alpha = 0.05; power = 0.80).

## Intervention

The interventions were performed on a side-alternating vibration platform (Galileo Fit, Novotec Medical, Germany) which was running at a frequency of 26 Hz (*Rittweger, Mutschelknauss & Felsenberg, 2003*; *Cochrane et al., 2010*) and off, respectively, for WBV and SHAM conditions. Subjects were instructed to maintain a half-squat position with their knees flexed at an angle of 60° (*Ritzmann et al., 2010*) for several sets with 2-minute rest between sets. Kinematics were controlled with a goniometer. The subjects stood with their feet 40 cm apart at a point where the tilting platform reached peak-to-peak displacement amplitude of five mm (*Ritzmann, Gollhofer & Kramer, 2013*).

At the beginning of each session, subjects underwent a 6-minute warm-up routine consisting of bench stepping (20 cm high) at a frequency of 0.5 Hz, swapping the leading leg at one minute intervals.

## Testing protocols

During the neuromuscular assessment, the subjects remained seated in a custom-built isometric knee extension apparatus equipped with a force transducer (MES, Maribor, Slovenia) (*Tomazin, Dolenec & Strojnik, 2008*; *García-Ramos et al., 2016*). The force transducer was calibrated prior to testing sessions. Each subject was seated in an upright position, hip at 100° and trunk leaning against the backrest of the testing apparatus, fixed by straps over the pelvis and a horizontal pad over the distal third of the thigh. The knee joint axis was aligned with the mechanical axis of the dynamometer. The shin pad was placed just superior to the medial malleolus. The right knee joint was fixed at a 60° angle (0° = full extension) (Fig. 1C).

## Femoral nerve electrical stimulation

The femoral nerve was stimulated by pressing a monopolar cathode (10-mm in diameter, Ag–AgCl, Type 0601000402, Controle Graphique Medical, Brie-Comte-Robert, France) into the femoral triangle of the iliac fossa (Fig. 1C). A larger (102 mm × 52 mm, Compex,

SA, Ecublens, Switzerland) self-adhesive electrode placed over the gluteal fold served as the anode. Electrical impulses (single, square wave, 1-ms duration) elicited by a high voltage constant current electrical stimulator (DS7A; Digitimer, Hertfordshire, UK) were used to trigger the muscle response, which was detected as a change in torque of the knee extensors. The stimulation intensity to elicit maximum knee extensors isometric twitch was determined in each subject at the beginning of each trial and maintained for the entire trial. Starting from an intensity of 10 mA, the stimulation intensity was progressively increased by 10 mA until no further increase in torque was observed despite further increment in electrical current. The current at maximal twitch torque was additionally increased by a factor of 1.5 to obtain a supramaximal stimulus (*Verges et al., 2009*).

### Single twitch

The torque change induced by a single supramaximal femoral nerve stimulus (*Place et al., 2007*) was analysed to obtain the peak torque value ($TW_{PT}$).

### High- and low-frequency doublets

The torque change induced by the paired high-frequency (100 Hz, i.e., 10-ms interstimuli interval) and low-frequency (20 Hz, i.e., 50-ms interstimuli interval) supramaximal electrical stimuli (*Place et al., 2007*; *Verges et al., 2009*) was analysed to obtain the following parameters: peak torque from 100 Hz doublet ($T_{100}$), peak torque from 20 Hz doublet ($T_{20}$). In addition, the low- to high-frequency ratio ($T_{20/100}$) was calculated using the following formula:

$$T_{20/100} = \frac{T_{20}}{T_{100}} * 100.$$

This ratio was used as a surrogate of low- to high-frequency tetanic stimulation (*Verges et al., 2009*).

### Maximal voluntary contraction with double twitch interpolated techniques

Subjects were asked to perform a 5 s maximal isometric voluntary knee extension (*Verges et al., 2009*). The signal was smoothed using a 0.5 s window moving average filter and peak torque (MVC) was retained for analysis. The double twitch interpolated technique (*Allen, Gandevia & McKenzie, 1995*) was performed by superimposing a 100 Hz doublet on the isometric plateau ($T_{MVC}$). A second analogous stimulation ($T_{100}$) on the relaxed muscle followed after 3 s (Fig. 1B). The ratio of the amplitude of the $T_{MVC}$ over $T_{100}$ was then calculated to obtain the level of voluntary activation (%VA):

$$\%VA = \left(1 - \frac{T_{MVC} - MVC}{T_{100}}\right) * 100.$$

### Statistics

A two-way factorial ANOVA (Type III) was conducted in R(3.5.1) with the afex package (*Singmann et al., 2018*) to compare the main effects of time ($t_0$, $t_f$, $t_{f15}$, $t_{f30}$) and trial

(WBV$_{30}$, WBV$_{60}$, WBV$_{180}$, SHAM$_{30}$, SHAM$_{60}$, SHAM$_{180}$) and the interaction effect of time $\times$ trial. Generalised eta squared ($\eta_G^2$) effect sizes were calculated for the ANOVA main and interaction effects. In the case of statistically significant interactions, post hoc comparisons with Sidak corrections were applied using the emmeans package (*Lenth et al., 2018*) in order to compare WBV and SHAM condition. Tukey-corrected pairwise post hoc tests were used to calculate differences to baseline within trials.

In addition to inference statistics, standardised changes in the mean of each measure were used to assess the magnitudes of effect (ES) between WBV and SHAM conditions of the same set duration (e.g., SHAM$_{30}$-WBV$_{30}$, SHAM$_{60}$-WBV$_{60}$, SHAM$_{180}$-WBV$_{180}$) and were then calculated using Cohen d. The magnitude of ES was interpreted as follows: trivial = <0.20; small = 0.2–0.59; moderate = 0.60–1.19; large = 1.20–1.99; and very large = >2.0 based on recommendations by *Hopkins et al. (2009)*.

Statistical significance was set at the level of $p < 0.05$. ES results should be interpreted with caution, since negative values imply a larger fatiguing effect of WBV compared to SHAM condition and positive values imply a larger fatiguing effect for SHAM condition compared to WBV.

## RESULTS

Descriptive statistics for MVC and %VA are displayed in Table 1; descriptive statistics for T$_{20}$, T$_{100}$ and T$_{20/100}$, TW$_{PT}$ are listed in Table 2.

### Maximum voluntary contraction
There was a statistically significant time effect ($F(3, 27) = 24.40$, $p < 0.001$, $\eta_G^2 = 0.02$), but no significant trial effect ($F(5, 45) = 2.13$, $p = 0.08$, $\eta_G^2 = 0.01$) nor trial x time interaction effect ($F(15, 135) = 0.60$, $p = 0.87$, $\eta_G^2 = 0.002$) for MVC. Within-trial post hoc tests showed differences between baseline and post-assessments (Fig. 2A).

### Level of voluntary activation (%VA)
There was a statistically significant time ($F(3, 27) = 3.67$, $p = 0.024$, $\eta_G^2 = 0.02$) and trial ($F(5, 45) = 2.52$, $p = 0.042$, $\eta_G^2 = 0.08$) effect, but no trial $\times$time interaction ($F(15, 135) = 1,21$, $p = 0.26$, $\eta_G^2 = 0.03$) for %VA. Post hoc tests did not reveal significant differences between baseline and post-assessments (Fig. 2B).

### Peripheral fatigue
There was a significant time effect ($F(3, 27) = 64.43$, $p < 0.001$, $\eta_G^2 = 0.25$) for T$_{20}$. Trial effects ($F(5, 45) = 1.91$, $p = 0.11$, $\eta_G^2 = 0.03$) and trial x time interaction effects ($F(15, 135) = 0.90$, $p = 0.56$, $\eta_G^2 = 0.007$) remained statistically insignificant. Post hoc tests revealed significant differences between baseline and post-assessments for each of the trials (Fig. 3A, Table 3).

There was a significant *time* effect ($F(3, 27) = 60.33$, $p < 0.001$, $\eta_G^2 = 0.15$) for T$_{100}$. *Trial* effect ($F(5, 45) = 2.15$, $p = 0.07$, $\eta_G^2 = 0.03$) and *trial $\times$time interaction* effect ($F(15, 135) = 0.43$, $p = 0.97$, $\eta_G^2 = 0.002$) remained statistically insignificant. Post hoc tests revealed significant differences between baseline and post-assessments for each of the trials (Fig. 3B, Table 3).

Kalc et al. (2020), *PeerJ*, DOI 10.7717/peerj.10388
**Table 1 Descriptive statistics (mean and SD), within trial relative change from baseline and Cohen d effects size for MVC and %VA.**

| | $t_0$ | $t_f$ | | | $t_{f15}$ | | | $t_{f30}$ | | |
|---|---|---|---|---|---|---|---|---|---|---|
| | mean (SD) | mean (SD) | Δ % | Cohen $d$ [95% CI] | mean (SD) | Δ % | Cohen $d$ [95% CI] | mean (SD) | Δ % | Cohen $d$ [95% CI] |
| *Maximum voluntary contraction (MVC)* | | | | | | | | | | |
| $SHAM_{30}$ | 206.04 (67.09) | 190.31 (68.15) | −7.63 | −0.21 [−0.31, −0.11] | 188.93 (68.26) | −8.30 | −0.23 [−0.37, −0.09] | 191.15 (63.97) | −7.23 | −0.21 [−0.35, −0.06] |
| $WBV_{30}$ | 219.46 (64.63) | 191.82 (55.25) | −12.59 | −0.42 [−0.66, −0.17] | 193.06 (67.16) | −12.03 | −0.36 [−0.54, −0.18] | 192.23 (64.92) | −12.41 | −0.38 [−0.54, −0.23] |
| $SHAM_{60}$ | 215.61 (64.31) | 201.05 (63.94) | −6.75 | −0.21 [−0.43, 0.02] | 199.34 (61.18) | −7.55 | −0.23 [−0.44, −0.03] | 200.52 (62.06) | −7.00 | −0.22 [−0.45, 0.01] |
| $WBV_{60}$ | 209.75 (63.53) | 194.58 (56.78) | −7.23 | −0.23 [−0.46, 0.00] | 201.37 (57.87) | −3.99 | −0.12 [−0.34, 0.09] | 197.64 (56.89) | −5.77 | −0.18 [−0.38, 0.01] |
| $SHAM_{180}$ | 207.81 (62.38) | 186.06 (51.73) | −10.47 | −0.34 [−0.53, −0.16] | 189.68 (52.72) | −8.72 | −0.28 [−0.49, −0.08] | 188.45 (57.65) | −9.32 | −0.29 [−0.47, −0.11] |
| $WBV_{180}$ | 200.36 (62.85) | 175.58 (53.91) | −12.37 | −0.38 [−0.68, −0.09] | 185.85 (61.52) | −7.24 | −0.21 [−0.44, 0.02] | 177.76 (65.08) | −11.28 | −0.32 [−0.66, 0.02] |
| *Level of voluntary activation (%VA)* | | | | | | | | | | |
| $SHAM_{30}$ | 93.05 (3.00) | 89.94 (6.14) | −3.34 | −0.58 [−1.22, 0.05] | 89.57 (7.14) | −3.74 | −0.58 [−1.34, 0.19] | 90.12 (4.47) | −3.15 | −0.40 [−0.72, 0.52] |
| $WBV_{30}$ | 90.74 (3.98) | 89.47 (5.07) | −1.41 | −0.25 [−0.97, 0.46] | 89.49 (4.58) | −1.38 | −0.26 [−0.87, 0.34] | 91.85 (4.21) | 1.22 | 0.25 [−0.39, 0.88] |
| $SHAM_{60}$ | 89.27 (4.78) | 87.71 (6.00) | −1.75 | −0.26 [−0.78, 0.26] | 88.14 (5.32) | −1.27 | −0.20 [−0.74, 0.34] | 87.52 (5.39) | −1.96 | −0.31 [−0.85, 0.23] |
| $WBV_{60}$ | 90.20 (4.85) | 87.41 (5.96) | −3.09 | −0.31 [−0.70, 0.55] | 91.33 (4.29) | 1.26 | 0.17 [−0.59, 0.67] | 89.66 (4.41) | −0.59 | −0.03 [−0.64, 0.62] |
| $SHAM_{180}$ | 87.40 (6.82) | 84.36 (5.47) | −3.47 | −0.45 [−1.08, 0.19] | 88.73 (5.06) | 1.52 | 0.20 [−0.55, 0.95] | 87.45 (6.01) | 0.05 | 0.01 [−0.75, 0.77] |
| $WBV_{180}$ | 88.54 (5.36) | 87.48 (4.31) | −1.20 | −0.17 [−0.67, 0.59] | 86.00 (6.28) | −2.87 | −0.40 [−1.03, 0.24] | 87.43 (4.33) | −1.25 | −0.21 [−0.98, 0.57] |

**Notes.**

$t_0$, baseline; $t_f$, after intervention; $t_{f15}$, 15 minutes after intervention; $t_{f30}$, 30 minutes after intervention.

**Table 2** Descriptive statistics (mean and SD), within trial relative change from baseline and Cohen d effects size for $T_{20}$, $T_{100}$, $T_{20/100}$ and TWPT.

| | $t_0$ | $t_f$ | | | $t_{f15}$ | | | $t_{f30}$ | | |
|---|---|---|---|---|---|---|---|---|---|---|
| | mean (SD) | mean (SD) | Δ % | Cohen d [95% CI] | mean (SD) | Δ % | Cohen d [95% CI] | mean (SD) | Δ % | Cohen d [95% CI] |
| **Low-frequency doublet ($T_{20}$)** | | | | | | | | | | |
| $SHAM_{30}$ | 75.26 (18.37) | 57.14 (12.56) | −24.07 | −1.04 [−1.51, −0.58] | 56.97 (11.84) | −24.30 | −0.63 [−0.78, 0.42] | 57.87 (10.33) | −23.11 | −0.63 [−0.78, 0.42] |
| $WBV_{30}$ | 80.47 (20.30) | 59.70 (13.91) | −25.80 | −1.08 [−1.43, −0.73] | 59.47 (15.47) | −26.10 | −1.05 [−1.39, −0.72] | 60.67 (15.04) | −24.61 | −0.63 [−0.78, 0.42] |
| $SHAM_{60}$ | 82.03 (22.79) | 62.84 (17.27) | −23.39 | −0.86 [−1.45, −0.27] | 66.78 (19.18) | −18.59 | −0.66 [−1.34, 0.03] | 62.78 (15.23) | −23.46 | −0.90 [−1.39, −0.41] |
| $WBV_{60}$ | 76.11 (18.34) | 56.88 (17.20) | −25.26 | −0.98 [−1.44, −0.52] | 57.06 (17.68) | −25.03 | −0.96 [−1.34, −0.58] | 55.79 (17.33) | −26.70 | −1.03 [−1.52, −0.54] |
| $SHAM_{180}$ | 82.24 (17.44) | 60.38 (12.87) | −26.58 | −0.63 [−0.78, 0.42] | 60.35 (13.30) | −26.62 | −0.63 [−0.78, 0.42] | 60.86 (13.15) | −26.00 | −0.63 [−0.78, 0.42] |
| $WBV_{180}$ | 79.59 (18.84) | 53.86 (16.14) | −32.33 | −1.33 [−1.69, −0.97] | 53.62 (17.56) | −32.63 | −1.29 [−1.68, −0.90] | 53.66 (13.88) | −32.58 | −1.42 [−1.82, −1.02] |
| **High-frequency doublet ($T_{100}$)** | | | | | | | | | | |
| $SHAM_{30}$ | 78.56 (20.46) | 63.90 (15.30) | −18.66 | −0.74 [−1.02, -0.45] | 63.09 (15.68) | −19.69 | −0.77 [−1.02, -0.52] | 62.73 (13.77) | −20.15 | −0.82 [−1.10, -0.54] |
| $WBV_{30}$ | 82.52 (21.65) | 67.91 (15.83) | −17.71 | −0.70 [−0.94, -0.45] | 65.79 (16.42) | −20.28 | −0.79 [−1.02, -0.56] | 66.10 (16.90) | −19.90 | −0.63 [−0.78, 0.42] |
| $SHAM_{60}$ | 87.98 (22.21) | 71.22 (18.06) | −19.05 | −0.75 [−1.04, -0.46] | 71.91 (20.34) | −18.27 | −0.68 [−1.04, -0.33] | 71.07 (18.78) | −19.23 | −0.75 [−1.08, -0.41] |
| $WBV_{60}$ | 81.00 (19.98) | 64.55 (20.02) | −20.30 | −0.75 [−1.32, -0.17] | 63.64 (19.06) | −21.43 | −0.81 [−1.36, -0.25] | 61.50 (18.58) | −24.06 | −0.92 [−1.57, -0.26] |
| $SHAM_{180}$ | 88.05 (21.36) | 69.93 (16.90) | −20.58 | −0.85 [−1.18, -0.52] | 68.09 (16.89) | −22.68 | −0.94 [−1.28, -0.60] | 67.69 (15.60) | −23.12 | −0.99 [−1.39, -0.59] |
| $WBV_{180}$ | 83.10 (21.07) | 63.93 (20.07) | −23.07 | −0.84 [−1.10, -0.59] | 62.99 (18.42) | −24.20 | −0.92 [−1.20, -0.64] | 62.06 (16.54) | −25.33 | −1.01 [−1.30, -0.71] |
| **Low- to high-frequency doublet ration ($T_{20/100}$)** | | | | | | | | | | |
| $SHAM_{30}$ | 0.96 (0.06) | 0.90 (0.07) | −6.56 | −0.89 [−1.45, -0.32] | 0.91 (0.08) | −5.15 | −0.68 [−1.09, -0.27] | 0.93 (0.08) | −3.27 | −0.43 [−0.78, -0.07] |
| $WBV_{30}$ | 0.97 (0.05) | 0.88 (0.05) | −10.11 | −0.60 [−0.77, 0.43] | 0.90 (0.07) | −7.50 | −0.49 [−0.74, 0.48] | 0.92 (0.07) | −5.77 | −0.44 [−0.73, 0.50] |
| $SHAM_{60}$ | 0.97 (0.07) | 0.88 (0.09) | −9.24 | −0.54 [−0.75, 0.46] | 0.93 (0.08) | −4.27 | −0.42 [−0.73, 0.51] | 0.94 (0.09) | −3.87 | −0.35 [−0.71, 0.53] |
| $WBV_{60}$ | 0.98 (0.06) | 0.89 (0.04) | −9.58 | −1.56 [−2.23, -0.89] | 0.90 (0.06) | −8.42 | −1.22 [−1.86, -0.57] | 0.91 (0.08) | −7.24 | −0.90 [−1.78, -0.02] |
| $SHAM_{180}$ | 0.96 (0.08) | 0.88 (0.08) | −7.64 | −0.83 [−1.28, -0.38] | 0.91 (0.09) | −4.98 | −0.49 [−0.82, -0.15] | 0.92 (0.08) | −4.15 | −0.44 [−0.80, -0.08] |
| $WBV_{180}$ | 0.98 (0.06) | 0.87 (0.12) | −11.12 | −1.00 [−1.59, -0.40] | 0.88 (0.07) | −9.53 | −1.28 [−1.80, -0.75] | 0.88 (0.09) | −9.78 | −1.10 [−1.54, -0.67] |
| **Single twitch peak torque ($TW_{PT}$)** | | | | | | | | | | |
| $SHAM_{30}$ | 26.81 (7.53) | 20.62 (6.21) | −23.10 | −0.63 [−0.78, 0.42] | 20.04 (5.12) | −25.28 | −0.63 [−0.78, 0.42] | 20.74 (5.39) | −22.65 | −0.84 [−1.29, −0.40] |
| $WBV_{30}$ | 26.62 (8.07) | 19.50 (5.65) | −26.77 | −0.63 [−0.78, 0.42] | 19.00 (5.53) | −28.64 | −1.00 [−1.36, −0.64] | 18.85 (5.34) | −29.19 | −1.03 [−1.42, −0.64] |
| $SHAM_{60}$ | 27.37 (8.27) | 19.83 (4.11) | −27.54 | −1.05 [−1.53, −0.56] | 19.06 (3.70) | −30.35 | −1.18 [−1.81, −0.54] | 19.41 (4.35) | −29.10 | −1.09 [−1.58, −0.60] |
| $WBV_{60}$ | 26.77 (8.11) | 19.76 (5.50) | −26.17 | −0.92 [−1.26, −0.57] | 18.25 (5.31) | −31.84 | −0.63 [−0.78, 0.42] | 19.71 (5.78) | −26.37 | −0.91 [−1.38, −0.44] |
| $SHAM_{180}$ | 26.72 (7.68) | 19.85 (6.47) | −25.72 | −0.88 [−1.09, −0.67] | 18.90 (5.35) | −29.29 | −1.07 [−1.36, −0.78] | 18.94 (6.39) | −29.15 | −0.63 [−0.78, 0.42] |
| $WBV_{180}$ | 27.20 (8.23) | 18.06 (7.74) | −33.60 | −1.04 [−1.32, −0.75] | 16.85 (7.37) | −38.06 | −1.20 [−1.58, −0.82] | 17.95 (7.35) | −34.01 | −1.07 [−1.43, −0.72] |

**Notes.**

$t_0$, baseline; $t_f$, after intervention; $t_{f15}$, 15 minutes after intervention; $t_{f30}$, 30 minutes after intervention.

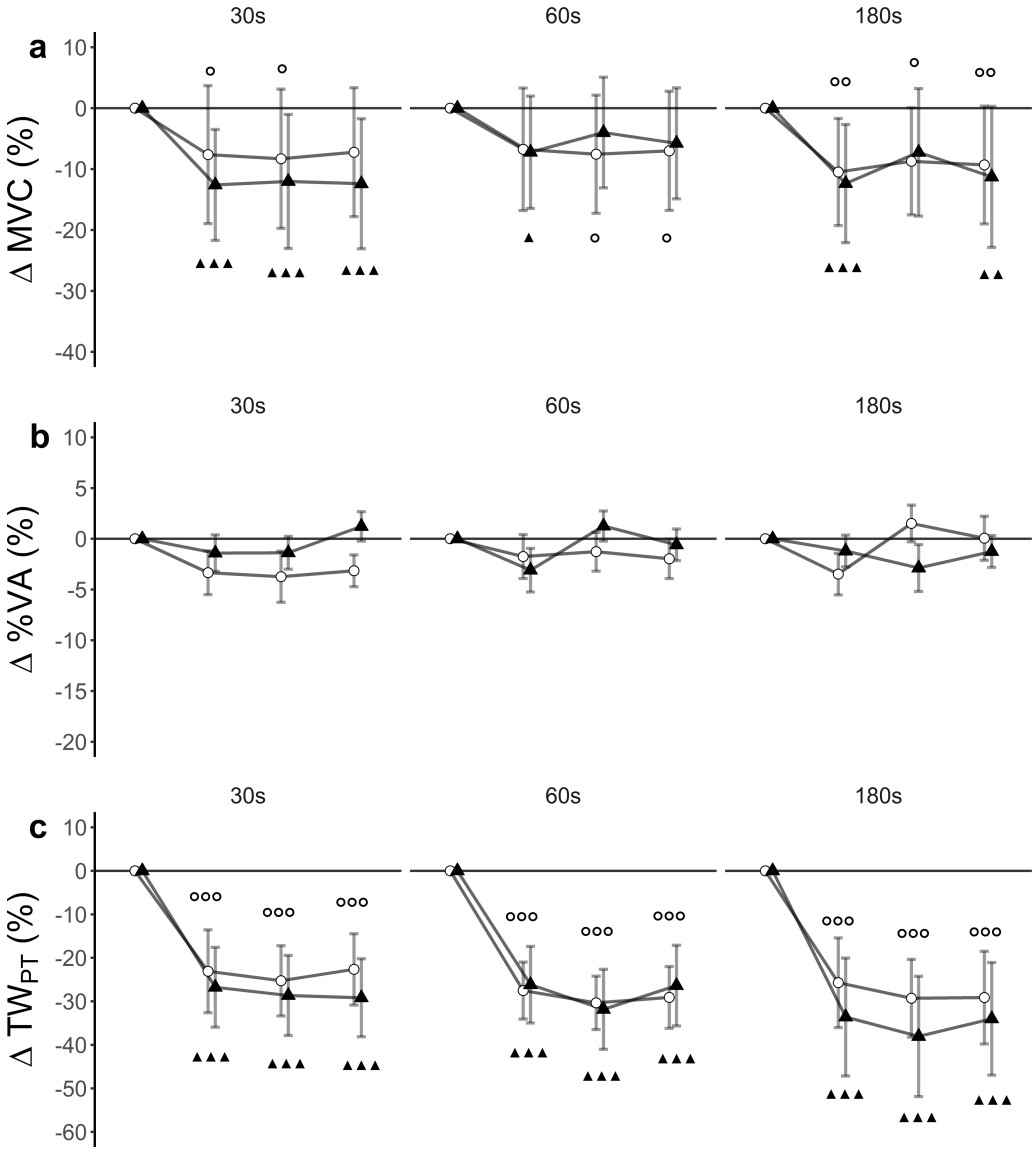

**Figure 2** **Relative changes from baseline.** (A) Maximum voluntary contraction (MVC), (B) level of voluntary activation (%VA) and (C) single twitch peak torque (TW$_{PT}$) for WBV (connected black triangles) and SHAM (connected white circles) for trials with different set durations (30 s, 60 s and 180 s). Values are expressed as mean and standard errors. Black triangles represent statistically significant WBV differences from baseline (▲▲▲ $p < 0.001$; ▲▲ $p < 0.01$; ▲ $< 0.05$). White circles represent statistically significant SHAM differences from baseline (○○○ $p < 0.001$; ○○ $p < 0.01$; ○ $p < 0.05$).

There was a significant time effect (F (3, 27) = 46.33, $p < 0.001$, $\eta_G^2 = 0.17$) for T20/100. Trial effect ($F(5, 45) = 1.06$, $p = 0.40$, $\eta_G^2 = 0.02$) and trial × time interaction effect ($F(15, 135) = 0.97$, $p = 0.49$, $\eta_G^2 = 0.02$) remained statistically insignificant. Post hoc tests revealed significant differences between baseline and post-assessments for each of the trials (Fig. 3C, Table 3).

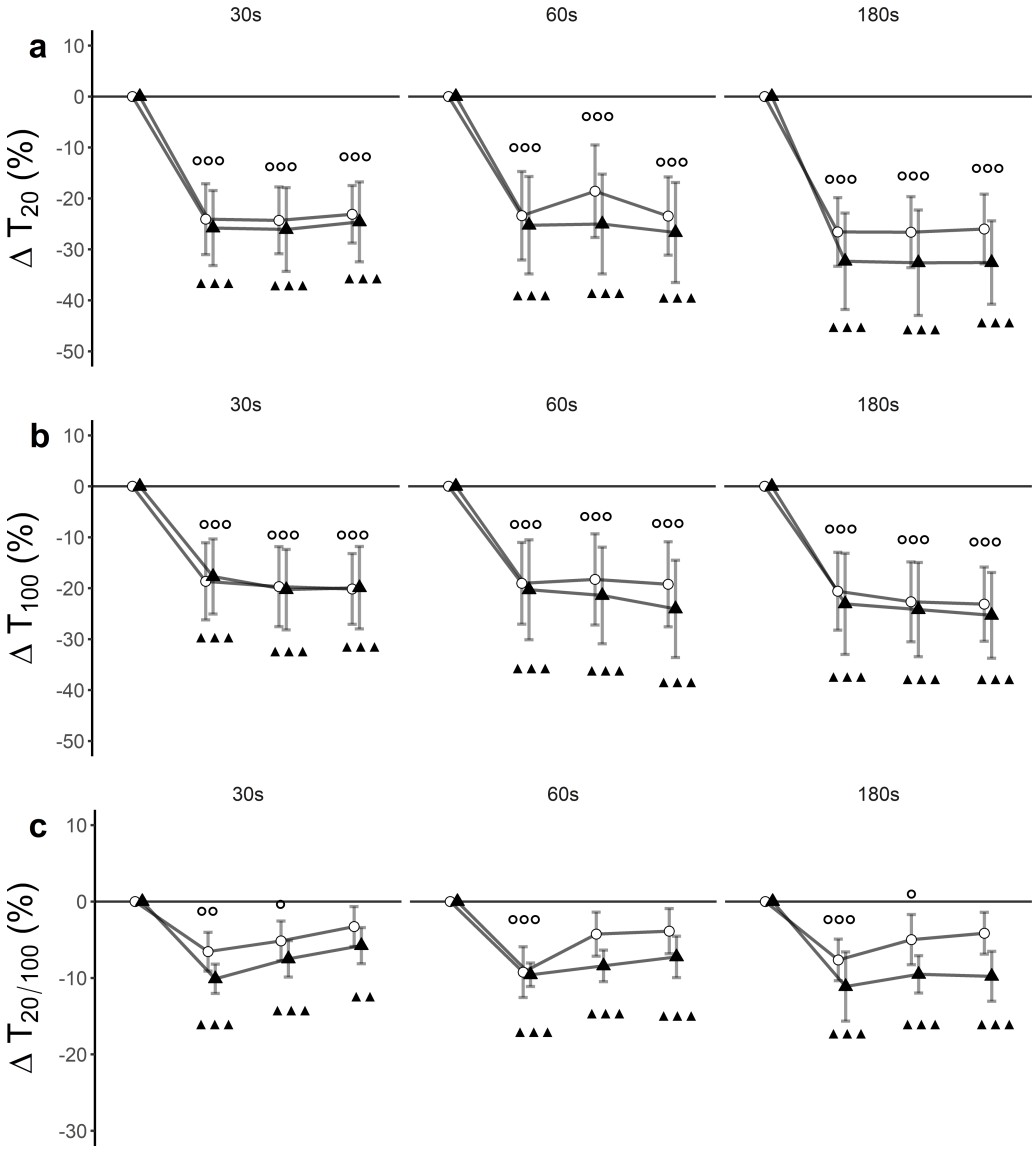

**Figure 3  Relative changes from baseline.** (A) low-frequency doublet ($T_{20}$), (B) high-frequency doublet ($T_{100}$) and (C) low-high torque frequency ratio ($T_{20/100}$) for WBV (connected black triangles) and SHAM (connected white circles) for trials with different set durations (30 s, 60 s and 180 s). Values are expressed as mean and standard errors. Black triangles represent statistically significant WBV differences from baseline (▲▲▲ $p < 0.001$; ▲▲ $p < 0.01$; ▲ $p < 0.05$). White circles represent statistically significant SHAM differences from baseline (○○○ $p < 0.001$; ○○ $p < 0.01$; ○ $p < 0.05$).

## Single twitch

There was a significant time effect (F $(3, 27) = 48.80$, $p < 0.001$, $\eta^2_G = 0.23$). Trial effects (F $(5, 45) = 0.86$, $p = 0.52$, $\eta^2_G = 0.006$) and trial x time interaction effect ($F(15, 135) = 1.05$, $p = 0.41$, $\eta^2_G = 0.006$) remained statistically insignificant for $TW_{PT}$. Post hoc tests revealed significant differences between baseline and post-assessments for each of the trials (Fig. 2C, Table 3).

Peerj

**Table 3** Within-trial Tukey corrected t-test comparison with baseline.

| | $t_f$ | | | $t_{f15}$ | | | $t_{f30}$ | | |
|---|---|---|---|---|---|---|---|---|---|
| visit | $t$-value | $p$-value | sig. | $t$-value | $p$-value | sig. | $t$-value | $p$-value | sig. |
| **Maximum voluntary contraction (MVC)** | | | | | | | | | |
| $SHAM_{30}$ | −2.53 | 0.04 | * | −2.75 | 0.02 | * | −2.40 | 0.05 | |
| $WBV_{30}$ | −4.45 | <0.001 | *** | −4.25 | <0.001 | *** | −4.38 | <0.001 | *** |
| $SHAM_{60}$ | −2.34 | 0.06 | | −2.62 | 0.03 | * | −2.43 | 0.05 | * |
| $WBV_{60}$ | −2.44 | 0.05 | * | −1.35 | 0.45 | | −1.95 | 0.15 | |
| $SHAM_{180}$ | −3.50 | 0.002 | ** | −2.92 | 0.01 | * | −3.12 | 0.006 | ** |
| $WBV_{180}$ | −3.99 | <0.001 | *** | −2.34 | 0.06 | | −3.64 | 0.001 | ** |
| **Level of voluntary activation (%VA)** | | | | | | | | | |
| $SHAM_{30}$ | −2.02 | 0.13 | | −2.26 | 0.07 | | −1.90 | 0.17 | |
| $WBV_{30}$ | −0.83 | 0.79 | | −0.81 | 0.80 | | 0.72 | 0.85 | |
| $SHAM_{60}$ | −1.01 | 0.68 | | −0.74 | 0.84 | | −1.14 | 0.59 | |
| $WBV_{60}$ | −1.81 | 0.20 | | 0.74 | 0.84 | | −0.35 | 0.98 | |
| $SHAM_{180}$ | −1.97 | 0.14 | | 0.86 | 0.77 | | 0.03 | 1.00 | |
| $WBV_{180}$ | −0.69 | 0.87 | | −1.65 | 0.27 | | −0.72 | 0.85 | |
| **Low-frequency doublet ($T_{20}$)** | | | | | | | | | |
| $SHAM_{30}$ | −6.08 | <0.001 | *** | −6.14 | <0.001 | *** | −5.84 | <0.001 | *** |
| $WBV_{30}$ | −6.97 | <0.001 | *** | −7.05 | <0.001 | *** | −6.65 | <0.001 | *** |
| $SHAM_{60}$ | −6.44 | <0.001 | *** | −5.12 | <0.001 | *** | −6.46 | <0.001 | *** |
| $WBV_{60}$ | −6.46 | <0.001 | *** | −6.40 | <0.001 | *** | −6.82 | <0.001 | *** |
| $SHAM_{180}$ | −7.34 | <0.001 | *** | −7.35 | <0.001 | *** | −7.18 | <0.001 | *** |
| $WBV_{180}$ | −8.64 | <0.001 | *** | −8.72 | <0.001 | *** | −8.71 | <0.001 | *** |
| **High-frequency doublet ($T_{100}$)** | | | | | | | | | |
| $SHAM_{30}$ | −5.42 | <0.001 | *** | −5.71 | <0.001 | *** | −5.85 | <0.001 | *** |
| $WBV_{30}$ | −5.40 | <0.001 | *** | −6.18 | <0.001 | *** | −6.07 | <0.001 | *** |
| $SHAM_{60}$ | −6.19 | <0.001 | *** | −5.94 | <0.001 | *** | −6.25 | <0.001 | *** |
| $WBV_{60}$ | −6.08 | <0.001 | *** | −6.41 | <0.001 | *** | −7.20 | <0.001 | *** |
| $SHAM_{180}$ | −6.70 | <0.001 | *** | −7.38 | <0.001 | *** | −7.52 | <0.001 | *** |
| $WBV_{180}$ | −7.08 | <0.001 | *** | −7.43 | <0.001 | *** | −7.78 | <0.001 | *** |
| **Low- to high-frequency doublet ration ($T_{20/100}$)** | | | | | | | | | |
| $SHAM_{30}$ | −3.57 | 0.001 | ** | −2.81 | 0.02 | * | −1.78 | 0.21 | |
| $WBV_{30}$ | −5.57 | <0.001 | *** | −4.13 | <0.001 | *** | −3.18 | 0.005 | ** |
| $SHAM_{60}$ | −5.09 | <0.001 | *** | −2.35 | 0.06 | | −2.13 | 0.10 | |
| $WBV_{60}$ | −5.30 | <0.001 | *** | −4.66 | <0.001 | *** | −4.01 | <0.001 | *** |
| $SHAM_{180}$ | −4.12 | <0.001 | *** | −2.69 | 0.02 | * | −2.24 | 0.08 | |
| $WBV_{180}$ | −6.14 | <0.001 | *** | −5.26 | <0.001 | *** | −5.40 | <0.001 | *** |

**Table 3** (*continued*)

| visit | $t_f$ | | | $t_{f15}$ | | | $t_{f30}$ | | |
|---|---|---|---|---|---|---|---|---|---|
| | $t$-value | $p$-value | sig. | $t$-value | $p$-value | sig. | $t$-value | $p$-value | sig. |
| *Single twitch peak torque (TW$_{PT}$)* | | | | | | | | | |
| SHAM$_{30}$ | −5.70 | <0.001 | *** | −6.24 | <0.001 | *** | −5.59 | <0.001 | *** |
| WBV$_{30}$ | −6.56 | <0.001 | *** | −7.01 | <0.001 | *** | −7.15 | <0.001 | *** |
| SHAM$_{60}$ | −6.93 | <0.001 | *** | −7.64 | <0.001 | *** | −7.33 | <0.001 | *** |
| WBV$_{60}$ | −6.44 | <0.001 | *** | −7.84 | <0.001 | *** | −6.49 | <0.001 | *** |
| SHAM$_{180}$ | −6.32 | <0.001 | *** | −7.20 | <0.001 | *** | −7.17 | <0.001 | *** |
| WBV$_{180}$ | −8.41 | <0.001 | *** | −9.52 | <0.001 | *** | −8.51 | <0.001 | *** |

**Notes.**

Asterisks represent statistically significant differences from baseline.

*** $p < 0.001$.

** $p < 0.01$.

* $p < 0.05$.

$t_0$, baseline; $t_f$, after intervention; $t_{f15}$, 15 minutes after intervention; $t_{f30}$, 30 minutes after intervention.

## DISCUSSION

The current study aimed to investigate the magnitude and origin of neuromuscular fatigue induced by long-exposure half-squat whole-body vibration intervention (WBV) with sets of different duration and compare it with non-vibration (SHAM) conditions. Our findings revealed no superimposed effect of WBV compared to control conditions without vibrations.

### Maximal voluntary contraction

Knee extensors MVC torque dropped by 7 to 12% after each fatiguing protocol, which is in line with other WBV induced fatigue studies, where MVC torque decreased by approximately 8% (*De Ruiter et al., 2003*; *Erskine et al., 2007*; *Colson et al., 2009*; *Zory et al., 2013*). Only *Maffiuletti et al. (2013)* reported a more substantial decline in MVC ($-23\%$), which is likely associated with the application of additional loads coupled with shorter inter-set rest periods compared to other studies and to our specific experimental setting. This finding is in contrast with our hypothesis that longer set duration exercises superimposed with vibration ($WBV_{180}$) would produce greater fatigue compared to $SHAM_{180}$ condition. However, it has been previously suggested that potentiated electrically elicited supramaximal doublets represent a more suitable indicator of peripheral fatigue and contractile impairments compared to MVC torque (*Place et al., 2007*).

### Central fatigue

The level of voluntary activation (%VA) of the knee extensors was not significantly depressed by any intervention utilised in this study, which suggest that mechanisms located in the central nervous system (CNS) were not significantly involved in the decline of MVC. These findings are in line with *Colson et al. (2009)* and *Jordan et al. (2010)* but in contrast to *De Ruiter et al. (2003)* who reported a vibration-induced decline in knee extensors voluntary activation. Despite *De Ruiter et al. (2003)* reported a similar drop in %VA compared to the present study (approx. 4%), any difference in interpretation between the two studies could be biased by the lack of a control group or control condition in *De Ruiter et al. (2003)* experiment coupled with the eligibility criteria for volunteers: in our study the population consisted of recreationally trained athletes and *De Ruiter et al.*'s (*2003*) enrolled untrained students. Evidence exist for physiological differences and perquisites in motor control between sedentary and trained active subjects (*Buford & Manini, 2010*). Being hypothesis-driven, our findings indicate that there are no evident superimposed effects of WBV on central fatiguing mechanisms compared to control conditions without WBV. This should be taken into consideration when designing exercise programs or research studies which intend to induce central fatigue. As such, WBV superimposed exercises are unlikely to be more effective than maintaining a static squat alone.

### Peripheral fatigue

To the best of our knowledge, this is the first study where electrically elicited supramaximal low- and high-frequency doublets were used to assess the origin and magnitude of peripheral fatigue after WBV exposure. For all protocols, $T_{20}$ was more affected than

$T_{100}$ leading to a decreased $T_{20/100}$ ratio (Fig. 3C). These declines suggest the occurrence of low-frequency fatigue (LFF) in all trials. It is noteworthy that the $T_{20/100}$ ratio for SHAM interventions returned to baseline values 15 min after the intervention, while WBV interventions remained significantly depressed up to 30 min after the intervention. This suggests that LFF is stronger and more long-lasting when a WBV exercise is executed with an emphasis on exposing to longer sets of vibration. The observation favouring LFF as an underlying mechanism can additionally be supported by the findings obtained from single twitch data. Similar to $T_{20}$ and $T_{100}$, $TW_{PT}$ progressively decreased as the intervention continued.

## Underlying mechanisms

The lack of difference between WBV and SHAM conditions observed in this study suggest that no beneficial effects on neuromuscular function exist when using superimposed WBV. This is particularly true for MVC and the level of voluntary activation. Even though some studies reported that WBV can induce modulation in the neuronal control, which is manifested as a facilitated central drive (*Mileva, Bowtell & Kossev, 2009*; *Krause et al., 2016*) this does not translate into central fatigue. Furthermore, the decline in low- and high-frequency doublets, as well as single twitch torque, suggests that a mechanism underlying the decrease in force production in both WBV and SHAM treatments is an impairment in $Ca^{2+}$ handling. This is followed by a gradual recovery of the $Ca^{2+}$ depletion within the 15–30 min following WBV equal to the SHAM intervention. Underlying cellular fatiguing mechanisms explaining the results for SHAM and WBV may refer to three aspects (*Westerblad et al., 2000*; *Allen & Westerblad, 2001*; *Williams & Ratel, 2009*): (a) since doublet peak torques progressively dropped at low- and high-frequencies of stimulation, there could be direct inhibition of inorganic phosphates ($P_i$) on $Ca^{2+}$, thereby producing an impairment in the cross-bridge force generation (*Millar & Homsher, 1990*). However, it is unlikely that this mechanism alone accounts for low-frequency fatigue (*Allen, Lannergren & Westerblad, 1995*). (b) It is likely that the larger drop in $T_{20}$ compared to $T_{100}$ could indicate a precipitation in $Ca^{2+}$-$P_i$ in the sarcoplasmic reticulum, leading to a decrease in free $Ca^{2+}$ available for release (*Allen & Westerblad, 2001*). In addition, (c) reduced myofibrillar $Ca^{2+}$ sensitivity can also affect force production (*Bruton et al., 2008*). Both mechanisms (b and c) have little impact on force production at high frequencies but a large effect on low frequencies (*Westerblad et al., 2000*).

## Limitations

The study might have some limitations. An important limitation of this study (similar to the majority of other vibration studies) is the lack of WBV load normalisation, as this may have considerable side-effects on the results, as was demonstrated by *Di Giminiani et al. (2009)*. Another limiting aspect deals with different work/rest ratios between long sets (180 s work −120 s rest) compared to other shorter set durations (30 s −120 s and 60 s −120 s). There is a great diversity in scientific and practitioner protocols and therefore, future studies should consider the variability in work/rest ratios and duration sets within the experimental design.

Furthermore, the experiment was executed in the right leg only. The leg dominance has thereby not been considered as a variable of influence on fatigue and fatigue mechanisms.

## CONCLUSIONS

The outcomes of this study suggest the origin of fatigue induced by half-squat with superimposed vibrations is no different from the control conditions without vibrations. Due to a lack of significant modulation of voluntary activation, it can be assumed that the fatiguing protocols used in this study predominantly affected peripheral mechanisms rather than central ones. The primary induced peripheral fatiguing mechanism seems to find its origin in low-frequency fatigue which most probably involves $Ca^{2+}$ handling. The outcomes of this investigation seems to suggest that static squat with superimposed whole-body vibrations does not represent a larger fatiguing stimulus compared to static squat alone in recreationally active athletes.

## ACKNOWLEDGEMENTS

We would like to thank all the participants who took part in the study.

### Funding
The project was funded by Human Resources Development in Sport 2016-2022, Slovenia and supported by the Slovenian Research Agency: ARRS / P5-0142-0381, Slovenia. The funders had no role in study design, data collection and analysis, decision to publish, or preparation of the manuscript.

### Grant Disclosures
The following grant information was disclosed by the authors:
Human Resources Development in Sport 2016-2022.
Slovenian Research Agency: ARRS / P5-0142-0381, Slovenia.

### Competing Interests
Ramona Ritzmann is employed by Praxisklinik Rennbahn AG. The authors declare there are no competing interests.

### Author Contributions
- Miloš Kalc conceived and designed the experiments, performed the experiments, analyzed the data, prepared figures and/or tables, authored or reviewed drafts of the paper, and approved the final draft.
- Ramona Ritzmann analyzed the data, authored or reviewed drafts of the paper, and approved the final draft.
- Vojko Strojnik conceived and designed the experiments, analyzed the data, authored or reviewed drafts of the paper, and approved the final draft.

## Human Ethics

The following information was supplied relating to ethical approvals (i.e., approving body and any reference numbers):

The University of Ljubljana, Faculty of Sport granted Ethical approval to carry out the study within its facility (Ethical Approval Ref: 975/2017).

## Data Availability

Data is available at Open Science Framework: Kalc, Milos. 2020. ''Data.'' OSF. November 6. doi:10.17605/OSF.IO/U94FK.

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
