# Peer review of "Effects of whole-body vibrations on neuromuscular fatigue: a study with sets of different durations"

_PeerJ, doi:10.7717/peerj.10388_

## Round 0.1 · original submission · Major Revisions

Reviewers generally provided positive feedback regarding the manuscript. However, some changes and clarifications are needed following the specific comments made by the reviewers.

Reviewer 1 ·

Basic reporting

The manuscript is well written overall.
I only have few comments.

Background in the abstract - increasing evidence ...induce a decline of neuromuscular parameters. I could be improved with an acute impairment in neuromuscular function

Introduction , line 34. Not sure about the use of reflectors here. Check english

Lines 48-51. The authors should also mention the differences in protocols and subjects with De Ruiter's findings (in their experiment they had untrained students) but in Maffiuletti's work there was external load added. I suggest the authors clarify this aspect and/or only cite literature with similar protocols and subjects to make their case

Experimental design

The cross-over design is appropriate, the experiment is well described and written so it can be replicated.

The statistical analysis as well, however I am not sure why the authors decided to add the magnitude based inferences approach. I think this makes the findings confusing. I suggest the authors to report the results only in the context of the two-way factorial ANOVA reported and discuss the effect sizes (eta squared) to interpret the magnitude of change rather than adding the magnitude of effects.

Validity of the findings

It is clear that there is no difference between trials and trial x time interaction and therefore the discussion should be focused on that aspect.

What is the biological meaning of a likely small to possibly small fatiguing effect on single twitch?

The comparisons with the results on central fatigue with De Ruiter's work should be discussed possibly due to the difference in the subjects used (the subjects in this study were recreationally trained athletes vs students untrained in De Ruiter's work).

Underlying mechanisms. Improve the first sentence. Probably the authors mean that there seems to be no beneficial effects of using WBV on neuromuscular function due to the lack of difference observed?

Lines 336-339 are a bit confusing, this sentence needs improving it is not clear what the authors are trying to say.

Line 340. Poor English. Are the authors attributing the observed acute decrease in force production in both treatments to an impairment in Ca2+ handling?
line 342- Following the intervention.

The following sentences are a bit too long and have few improvements needed. Also this part should be discussed in the context of the findings. The way it is currently written it is assuming that the intervention (WBV) caused an impairment in neuromuscular function, while the statistics analysis reveals no difference with the sham intervention. So the authors should clarify the last part better and make clear that there was no difference and therefore, the observed reduction over time is possibly caused by the same mechanisms of the sham intervention?

Limitations
While the authors explain well the limitations of the study. I am not sure if the work/rest ratios is an actual limitation. The study was designed for this scope and therefore it was a design choice? I am surprised as well that there was no larger evidence of fatigue with this protocol (180s work - 120s rest).

Conclusions.
As indicated above, results should be discussed in light of the statistical findings and not with the MBI. I have the impression that at all costs there is the willingness to identify a positive aspect when the data are clearly showing the opposite. I am confident the authors can address this properly as negative findings are as important as positive ones.

Also, I am am a bit puzzled of how the findings can be used by practitioners. Since there is no difference between treatments on the acute measures of neuromuscular function, what is the advantage of using vibration with the protocols and equipment used? This should go in the conclusion

My first impression is that probably the sample size was too small despite the a-priori power calculation (Jordan's study used 24 subjects) and they only found a difference in voluntary muscle activation. On what parameter was the power analysis conducted for the Jordan study?

Reviewer 2 ·

Basic reporting

The study is very clear, well written.

Experimental design

No comments.

Validity of the findings

Studies show that more than 10 minutes of WBVE can cause fatigue; why do you use 9 minutes ?

Additional comments

Congratulations. This study is very important to our comunity. We need to know about WBVE and fatigue.

·

Basic reporting

The reviewed paper adheres to all PeerJ policies and within the journal scope. The manuscript is written in the proper English style and conform to professional standards of courtesy and expression. The introduction is sufficient with relevant referenced literature. The structure of the article is typical for an original research article.

Experimental design

The authors define the clear research question, connected with the identified gap of knowledge, presented at the broad background in the introduction. Methods and protocols are described very well, also with very detailed statistical analysis. The measurements performed with high technical and ethical standards.

Minor comments:
Line: 104: „different visits with at least seven days rest in-between.” – please check it with Figure 1A, there is „separated by 6 days rest in between”.

Validity of the findings

The data have been provided, but some of them weren't compatible (I described this at the end in detail) and the figures should be revised. The results are enough to verify hypotheses and make conclusions. The discussion section is prepared nicely. Conclusions are well stated, linked to the research question. The authors have noted some limitations of the study.

Minor comments:
Line 270: „…statistically insignificant for TWPT”. – did you mean „…for TWPT”? {last two letters in subsricpt}

Line 325 and 326 – there is no explanation of ‘LFF’. Is it low-frequency fatigue? Add abbreviation in line 319 - should be enough.

Description of Figure 4: you put one black triangle with one white circle together as p<0,05.

I have one question for the research protocol: only right legs were measured, so are they all were right –legged persons? Did you check it? If yes, describe how; if not, please put some opinion about the lower limb lateralization and its influence on your results.

The other minor comment on data presentation at figures is about showing native units at graphical parts. It suggests that you present raw data or maybe raw differences for readers who didn’t read carefully the description of the figure. As you noted – the graphics show ES as d Cohen value, so that value hasn’t any unit. Please verify this for Figures 3 and 5. Please consider is it necessary to show the same both in numerical and graphical parts? What is ‘MBI’ abbreviation?

The major comment is about data compatibility. I compared the relative changes from baseline between Figure 2 and Table 1 for MVC and %VA. The graphical representations didn’t match numerical values. For example, MVC(%) at Figure 1A-30s set and connected black triangles for WBV. I clearly see that the relative change is bigger at tf15 then tf, and the biggest at tf30 but data from Table 1 are -12.59% at tf, -12.03 at tf15, and -12.41 at tf30; that’s why I’m confused. Look also for %VA where for every set (30, 60, and 180s) I found differences. For example (Fig. 1B, 180s set) relative difference at tf for WBV is a positive value (around 2-3% based on the graph) but Table 1 showed -1.20 value.
I didn't check all values at Figures 2 and 4 with connection to Tables 1 and 2 – now it’s your job to do it precisely.

Additional comments

Summarise my review – the paper is very good but necessary is a major revision on data compatibility between graphical and numerical representation. Other comments are minor.

---

## Round 0.2 · accepted · Accept

Congratulations for meeting the high standard publications of PeerJ.

·

Basic reporting

The reviewed paper adheres to all PeerJ policies and within the journal scope. The manuscript is written in the proper English style and conform to professional standards of courtesy and expression. The introduction is sufficient with relevant referenced literature. The structure of the article is typical for an original research article.

Experimental design

The authors define the clear research question, connected with the identified gap of knowledge, presented at the broad background in the introduction. Methods and protocols are described very well, also with very detailed statistical analysis. The measurements performed with high technical and ethical standards.

Validity of the findings

The data have been provided. The results are enough to verify hypotheses and make conclusions. The discussion section is prepared nicely. Conclusions are well stated, linked to the research question. The authors have noted some limitations of the study.

Additional comments

Summarise my review – the paper is very good. All previous major and minor recommendations (especially revision on data compatibility between graphical and numerical representation) were made.